# Genome-Wide Characterization and Phylogenetic Analysis of GSK Genes in Maize and Elucidation of Their General Role in Interaction with BZR1

**DOI:** 10.3390/ijms23158056

**Published:** 2022-07-22

**Authors:** Hui Li, Li Luo, Yayun Wang, Junjie Zhang, Yubi Huang

**Affiliations:** 1State Key Laboratory of Crop Gene Exploration and Utilization in Southwest China, Sichuan Agricultural University, Chengdu 611130, China; lihui8627@outlook.com (H.L.); 18728520454@sohu.com (L.L.); wangyayun3104@outlook.com (Y.W.); 2College of Agronomy, Sichuan Agricultural University, Chengdu 611130, China; 3College of Life Science, Sichuan Agricultural University, Ya’an 625014, China

**Keywords:** GSK-3, BRs, gene expression

## Abstract

Glycogen synthase kinase-3 (GSK-3) is a nonreceptor serine/threonine protein kinase that is involved in diverse processes, including cell development, photomorphogenesis, biotic and abiotic stress responses, and hormone signaling. In contrast with the deeply researched GSK family in *Arabidopsis* and rice, maize GSKs’ common bioinformatic features and protein functions are poorly understood. In this study, we identified 11 *GSK* genes in the maize (*Zea mays* L.) genome via homologous alignment, which we named Zeama;GSKs (ZmGSKs). The results of ZmGSK protein sequences, conserved motifs, and gene structures showed high similarities with each other. The phylogenetic analyses showed that a total of 11 genes from maize were divided into four clades. Furthermore, semi-quantitative RT-PCR analysis of the *GSKs* genes showed that ZmGSK1, ZmGSK2, ZmGSK4, ZmGSK5, ZmGSK8, ZmGSK9, ZmGSK10, and ZmGSK11 were expressed in all tissues; ZmGSK3, ZmGSK6, and ZmGSK7 were expressed in a specific organization. In addition, *GSK* expression profiles under hormone treatments demonstrated that the *ZmGSK* genes were induced under BR conditions, except for *ZmGSK2* and *ZmGSK5*. *ZmGSK* genes were regulated under ABA conditions, except for *ZmGSK1* and *ZmGSK8*. Finally, using the yeast two-hybrid and BiFC assay, we determined that clads II (ZmGSK1, ZmGSK4, ZmGSK7, ZmGSK8, and ZmGSK11) could interact with ZmBZR1. The results suggest that clade II of ZmGSKs is important for BR signaling and that ZmGSK1 may play a dominant role in BR signaling as the counterpart to BIN2. This study provides a foundation for the further study of GSK3 functions and could be helpful in devising strategies for improving maize.

## 1. Introduction

Glycogen synthase kinase-3 (GSK-3) proteins constitute an important group of kinases. GSK-3 regulates fundamental biological pathways through inhibition and lies downstream of multiple cell signaling factors in eukaryotes [1,2,3]. GSK-3 kinases are defined by a conserved serine/threonine protein kinase catalytic domain (PF0069.24) [4]. GSK-3 was first reported in mammals as a kinase [5], and two isoforms have been reported in mammals: GSK-3α and GSK-3β. Both are involved in multiple physiological processes, including the regulation of the cell cycle, the induction of cell differentiation, and glycogen metabolism [6,7]. In contrast, *GSK-3* kinase genes have been shown to have more diverse isoforms and functions in plants [1,8]. However, the isoform composition and functions of ZmGSK-3 proteins in maize are unclear. Notably, whether they are involved in hormone signaling is unknown.

BIN2 is a member of the GSK-3 kinase family that has been identified in many plants [9,10]. In Arabidopsis, BIN2 participates in the regulation of stress response by interacting with and phosphorylating RD26 [11]. *TaGSK3*, the wheat ortholog of Arabidopsis *BIN2*, induces round grains [12]. In addition, the knockout and overexpression of *OsGSK1*, an ortholog of Arabidopsis *BIN2*, showed that *OsGAK1* plays important roles in the stress-response signaling pathway and reproductive organ development [13]. In Arabidopsis, a GSK-3 kinase mutant named *dwf12* exhibits significant dwarfism and leaf curling [14]. *GSK-3* knockouts exhibit morphological similarities to Brassinosteroid (BR)-signaling mutants and display insensitivity to brassionazole (BRZ; an inhibitor of BR biosynthesis) [15]. Similar to the BR-signaling mutant, other *GSK-3* mutant or overexpression models have indicated a negative role of GSK-3 in the BR signaling pathway, which led to its being recognized as a phytohormone.

Numerous studies have revealed that plant GSK-3 kinases generally function through interaction with transcription factors, especially BZR1 [16]. For instance, BIN2 mainly regulates downstream BR-related transcription networks by controlling the phosphorylation status of BZR1 to mediate the BR signaling pathway in Arabidopsis [17]. In the absence of BRs, BZR1/BES1 function can be inhibited due to phosphorylation by BIN2 [18,19]. In the presence of BRs, BZR1/BES1 dephosphorylates and accumulates in the nucleus to regulate target-gene expression because BIN2 kinase activity is inhibited [20]. BZR1 transcription factors play critical roles in BR signaling [21]. BZR1 regulates CBF1 and CBF2 expression to modulate Arabidopsis’ freezing tolerance [22]. BZR1 can interact with other transcription factors to participate in BR-induced gene regulation. For example, AtIWS1 can interact with BES1 both in vitro and in vivo to promote plant-steroid-induced gene expression [23]. Another example is AtMYB30, an MYB transcription factor that cooperatively interacts with BES1 to promote BR-target-gene expression and is targeted by BES1 to amplify BR signaling [24]. In addition, BES1 can recruit other transcription factors to coordinate BR responses [25].

In contrast, little is known about whether the GSK-3 family could interact with BZR1 to participate in the BR signaling pathway in maize. Maize is the main cereal grain crop that contributes to solving the food security problem, and it is also known to be a social security concern for farmers [26,27,28,29]. Thus, understanding the GSK-3 family is significant. In this report, we aimed to investigate the role of ZmGSK in maize using a comprehensive analysis of phylogeny, gene structure, conserved motifs, *cis* elements, chromosomal location, and protein–protein interaction. Furthermore, we examined the expression pattern of *ZmGSK* genes in various developmental tissues, investigating their function using different hormones, and probing their interaction with BZR1.

## 2. Results

### 2.1. Genome-Wide Identification of ZmGSKs in Maize Genome

After removing redundant sequences, 11 GSK genes were identified in the maize genome. The parameters used to describe the ZmGSK proteins are listed in Table 1. Putative ZmGSK proteins varied from 403 (ZmGSK7) to 1076 (ZmGSK3) amino acids (AAs) in length, with molecular weights (*M*_W_s) ranging from 45.17 to 120.83 KDa. Additionally, the theoretical pI of most ZmGSK proteins was higher than 7, except for those of ZmGSK6 (6.79) and ZmGSK3 (6.33).

### 2.2. Comparative and Phylogenetic Analyses of ZmGSKs

In order to precisely reveal the evolutionary relationships of the ZmGSK proteins, we performed phylogenetic analyses on 30 GSK proteins, including 11 maize, 10 Arabidopsis, and 9 rice GSKs, using the neighbor-joining method (Figure 1). All proteins were categorized into four groups (I, II, III, and IV). Meanwhile, the number and protein sequences of ZmGSKs were more similar to those of OsGSKs than to those of AtASKs, indicating that GSKs from maize have a closer relationship with GSKs from rice than with those from Arabidopsis. This shows that the GSK gene is not highly conserved in the evolution of species and diversified during its evolution.

### 2.3. The Analyses of Chromosomal Locations, Gene Structures, and Conserved Motifs

The *ZmGSK* genes were unevenly distributed among the 10 chromosomes of the maize genome (Figure 2). The number of genes on each chromosome ranged from 0 (chromosomes 2 and 7) to 2 (chromosomes 3, 5, and 8), and the other five *ZmGSK* genes were evenly distributed in chromosomes 1, 4, 6, 9, and 10 (one gene on each chromosome).

The sequence alignment of the ZmGSK proteins is shown in Appendix A. The ~280-amino acid is near the carboxyl terminus. The phylogenetic tree branches of maize GSKs were categorized into four groups (Figure 3a). MEME was used to analyze conserved motifs in the 11 ZmGSK proteins (Figure 3b, Appendix A). In total, we identified 15 putative conserved motifs, with 6–50 residues and E-values < 1.2 × 10^−22^. Motifs 1–8 were found in almost all members of the ZmGSK family. All GSKs contained a highly conserved S_TKc domain (Appendix A). In addition, ZmGSK3 contained two KH domains and a few PPR domains. To gain insights into the structural features of the predicted ZmGSK genes, we determined their intron/exon distribution (Figure 3c). The structural analysis indicated that the coding regions of all maize GSK genes were interrupted by 10–12 introns. ZmGSK3 was an exception to this, with 19 introns in the gene. The UTR regions of ZmGSK3 were undetermined, while other maize GSK genes had UTR-region information in the genomic annotation files. Overall, highly similar gene structures were observed for the four groups of GSK family genes, especially for groups I, II, and III, according to the number and length of exons.

### 2.4. Cis-Acting Elements of ZmGSK Genes in Maize

To further understand how *ZmGSK* genes are regulated, we detected their upstream promoter sequences (2000 bp upstream of the coding region) and subjected them to cis-acting element prediction using PlantCARE. Twenty cis-acting elements, including abscisic-acid responsiveness (ABRE), low-temperature responsiveness (LTR), TGA elements (auxin responsiveness), MeJA responsiveness (CGTCA motif), circadian control (circadian), anaerobic induction (ARE), MYB binding site (MYB), anoxic-specific inducibility (GC motif), gibberellin-responsive element (P-box), wound responsiveness (WUN motif), meristem expression (CAT-box), MYC, defense and stress responsiveness (TC-rich repeats), salicylic-acid responsiveness (TCA), gibberellin-responsive elements (GARE motif), light responsiveness, gibberellin responsiveness (TATC-box), auxin responsiveness, MeJA responsiveness (TGACG motif), and the binding site of AT-rich DNA-binding protein (AT-rich element), were mapped onto the promoter regions (Figure 4). ABRE elements were found in nine *ZmGSK* genes promoters, including those of *ZmGSK2*, *-3*, *-4*, *-5*, *-7*, *-8*, *-9*, *-10*, and *-11*. ABRE elements showed more hormone responsiveness, and some *ZmGSK* gene promoters contained at least one ABRE element. MYB-binding sites were identified in all the promoters of *ZmGSK* genes. All the *cis* elements identified in this study are involved in hormone signaling, the stress response, and important transcription-factor-binding sites.

### 2.5. Expression Patterns of ZmGSK Genes in Different Tissues and Organs

To further define their distinct biological functions in development, the expression patterns of all *ZmGSKs* were analyzed using public RNA-seq data [30] (Figure 5a) and semiquantitative RT-PCR (Figure 5b). Our results showed that *ZmGSK9* was expressed in different tissues and that the expression was much higher in mature leaves, especially after pollination, indicating that its function is mainly related to source regulation. *ZmGSK5* was highly expressed in anthers, showing that this gene is involved in anther development. *ZmGSK4* and *ZmGSK11* exhibited ubiquitous expression in all tissues, but *ZmGSK4* was expressed at higher levels in the endosperm, as shown by RT-PCR. The transcript level of *ZmGSK8* was much higher in the cob and silk tissues than in other tissues. *ZmGSK1* and *ZmGSK3* expression were similar in roots, stems, and leaves. *ZmGSK1* was expressed at a higher level in silk and decreased with seed development, while *ZmGSK3* expression was increased with seed and endosperm development, suggesting that *ZmGSK1* is involved in fertilization and that *ZmGSK3* focuses on endosperm filling and maturation. However, *ZmGSK3* was detected at a low level in all tissues using RT-PCR. The transcript levels of *ZmGSK7* were much higher in silk and developing embryos. *ZmGSk6* and *ZmGSK10* exhibited similar expression patterns in the root, stem, leaf, cob, seed, and endosperm, but *ZmGSK10* had higher expression levels in the embryo. *ZmGSK2* was detected at a high level in the endosperm, while it was missing from the MaizeGDB database.

To investigate the response of these genes to BR and ABA starvation, we harvested Mo17 endosperms 9 or 10 DAP; grew them on normal, BR, and ABA 1/2 MS media for 48 h; and subjected them to qRT-PCR analysis. Our data showed that eight genes (*ZmGSK3*, *ZMGSK4*, *ZmGSK6*, *ZmGSK7*, *ZmGSK8*, *ZmGSK10*, and *ZmGSK11*) were downregulated after BR treatment for 48 h compared with after treatment with CK (Figure 5c). Meanwhile, *ZmGSK1* and *ZmGSK9* were upregulated by more than two-fold after BR treatment. Five genes (*ZmGSK4*, *ZmGSK5*, *ZmGSK6*, *ZmGSK9*, and *ZmGSK10*) were upregulated, and four genes (*ZmGSK2*, *ZmGSK3*, *ZmGSK7*, and *ZmGSK11*) were downregulated after ABA treatment, compared with after CK treatment.

### 2.6. Protein–Protein Interaction of ZmGSKs and BZR1

Previously, BZR1 was identified to interact with BIN2 and could be phosphorylated by BIN in Arabidopsis [31,32]. As BZR1 is a transcription factor and has to activate transcription to function, we investigated whether the fused BD-GSK3 protein interacted with AD-BZR1 using a yeast two-hybrid experiment. The results showed that ZmGSK1, ZmGSK3, ZmGSK4, ZmGSK7, ZmGSK8, and ZmGSK11 could interact with BZR1 (Figure 6a). The other ZmGSK proteins did not interact with BZR1. The ZmGSKs–ZmBZR1 interaction was further verified with bimolecular fluorescence complementation (BiFC) assays. ZmGSKs were fused to a C-terminal yellow fluorescent protein (YFP) fragment (ZmGSKs–cYFP), and the ZmBZR1 protein was fused to an N-terminal YFP fragment (ZmBZR1–cYFP). When fused ZmGSKs–cYFP was co-expressed with ZmBZR1–nYFP in onion, a YFP fluorescence signal was detected in transformed-cell nuclei (Figure 6b). No fluorescence was observed in negative-control experiments (Figure 6b). These results suggest that ZmGSKs kinase physically interacts with the ZmBZR1 transcription factor in plant-cell nuclei to participate in the BR pathway.

## 3. Discussion

The plant GSK3 protein family plays an important role in growth and development [33], and participates in biological- and abiotic-stress responses [34]. BIN2, a member of the GSK3 family, has been reported as a negative regulator of BR signaling in different species [10,35,36,37]. Our findings could accelerate the further functional characterization of *ZmGSK* genes and provide a framework to clarify the roles of these genes in biology and hormone response.

### 3.1. Classification and Phylogenetic Analysis of the Maize GSK Family

A number of *GSK* genes have been identified in Arabidopsis [4,38,39] and rice [13]. In contrast with mammals, GSK proteins are much more diverse in plants; for example, there are ten GSK isoforms in Arabidopsis [9] and nine in rice [40]. Here, we found eleven GSK genes in maize, which might reflect the larger size of the maize genome.

Our phylogenetic comparison of the GSK proteins included maize, rice, and Arabidopsis, and all 30 GSK genes were divided into four subgroups (Figure 1). The diverse subgroups of gene motifs were different, but there was a highly conserved motif common to all subgroups (Figure 3b). For example, subgroup I of ZmGSKs contained a conserved domain (motif 12) in the N terminus and motif 11 only existed in subgroup III. Meanwhile, the relationship of maize GSK genes in the two trees was comparable. For example, the branches of ZmGSK1, ZmGSK4, ZmGSK7, ZmGSK8, and ZmGSK11 in the maize NJ tree were also included in subgroup II in the total NJ tree. A genetic analysis of T-DNA insertional mutants of group II in Arabidopsis revealed that their function is redundant in BR signaling but that BIN2 plays a dominant role [15,41]. In addition, both OsGSK1 and OsGSK2 are orthologs of ASK21 (BIN2), whereas only OsGSK2 appeared to play a predominant role in BR signaling [35,42]. Thus, OsGSK2 is considered to be the rice counterpart of Arabidopsis BIN2 because it has the highest sequence similarity [42]. A previous study found that ZmASK1, named ZmGSK3 in our study, belongs to group IV in maize. However, BIN2 belongs to group II in rice and Arabidopsis. Thus, the maize counterpart of BIN2 remains to be identified. Here, we found that both ZmGSK1 and ZmGSK8 have a high sequence similarity with BIN2 of rice and Arabidopsis. Therefore, one of these two genes is likely maize BIN2. Our logical classification may be helpful in further studies of GSK protein function.

### 3.2. Diverse Expression Patterns and Hormone Responses Generate Differential Functions

Detailed analyses of gene expression patterns and hormone responses can provide considerable insight into the critical regulatory points in which they are implicated [43,44,45]. Based on the maizeGDB database and our semi-quantitative RT-PCR results, the expression patterns of the candidate GSK genes were differentiated. Most of the GSK genes in maize were expressed in various tissues during plant development, but some genes showed higher transcript levels in specific organs (Figure 5a). For example, *ZmGSK2*, *ZmGSK4*, and *ZmGSK7* were mainly expressed in the endosperm, whereas ZmGSK1, ZmGSK8, and ZmGSK10 were consistently weakly expressed in all analyzed organs. The observed expression patterns were consistent with a previous analysis of ZmASK1 [46]. Diverse expression patterns may imply differential functions. For example, AtSK11 and AtSK12 are mainly expressed during flower development and are required for the patterning of gynecium [38]. In an OsGSK1 knockout mutant, OsGSK1 expression was the highest in young panicles, and spikelets were larger [13]. In our study, the relatively high expression of maize GSK genes in reproductive organs was more obvious, indicating that these GSK genes are involved in the development of reproductive organs. Meanwhile, we found that some subgroups of the GSK gene showed similar expression patterns, such as ZmGSK1 and ZmGSK8, indicating that they perform similar functions. However, the expression of ZmGSK1 was higher than that of ZmGSK8. These data suggest that ZmGSK1 is more important than ZmGSK8 and may be the counterpart of BIN2 in maize. In addition, group I members showed diverse expression patterns, similar to Arabidopsis [4]. ZmGSK2 was mainly expressed in the endosperm, and ZmGSK5 was expressed in anthers, while ZmGSK9 was detected at high levels in leaves.

In our study, we focused on identifying the genes involved in seed development. Thus, we analyzed the expression of eleven genes under BR and ABA conditions. The quantitative results indicated that all the GSK genes were induced in seeds by BR (except for ZmGSK5) and that nine GSK genes were induced under ABA conditions (Figure 5b). Meanwhile, for most of the GSK genes, the expression levels were the opposite under BR and ABA conditions, and a high BR expression was accompanied by low expression under ABA, and vice versa. Although these results indicated that GSK genes have an antagonistic effect on BR and ABA, which regulate seed development, the effect of the BR and ABA signaling pathways is apparently complicated [47]. These results are consistent with previous research on seed germination [48,49]. In addition to studying expression, other studies are needed to clarify the biological functions of the isolated genes with respect to their diverse expression patterns and hormonal responses.

Previous studies on the function of GSK proteins mainly focused on interactions with BZR1 [20]. In our study, we found that members of groups II and IV in maize interacted with BZR1 (Figure 6a). A study on Arabidopsis also found that members of group II ASKs could bind to BZR1 as a negative regulator in BR signaling [50]. Meanwhile, members of group III ASKs in Arabidopsis interact with BEH2 [51]. Thus, it is suggested that members of groups II and IV mediate the hormonal pathway through interaction with BZR1 and that groups I and III may interact with other transcription factors to participate in hormonal responses. Previous studies showed that ZmBZR1could bind to the KRP6 and GRACE promoter to regulate seed size [52]. Here, we found that ZmGSK1, ZmGSK4, ZmGSK7, and ZmGSK8, which could interact with ZmBZR1, were expressed in the endosperm. These results show that ZmGSK1, ZmGSK4, ZmGSK7, and ZmGSK8 play important roles in seed development.

## 4. Methods and Materials

### 4.1. Plant Materials

Seeds from maize inbred line Mo17 were obtained from Maize Research Institute of Sichuan Agricultural University (latitude, 30.705207; altitude, 511 m). To analyze tissue-specific gene expression patterns in Mo17, plants were grown under standard crop-management conditions in Chengdu, China. Roots, stems, and leaves were collected at the shooting stage. Additionally, seeds, endosperms, and embryos were harvested 15 days after pollination (DAP). Anthers and silks were gathered during plant pollination. All collected samples were immediately frozen in liquid nitrogen and stored at −80 °C until use.

### 4.2. Identification of Maize GSK Genes

Putative *ZmGSK* genes were identified via homologous alignment. The *ZmGSK* family genes from *Oryza sativa* and *Arabidopsis thaliana* were downloaded from the Tair database (https://www.arabidopsis.org (accessed on 1 July 2019) and the rice annotation project database (http://rapdb.dna.affrc.go.jp/download/irgsp1.html (accessed on 1 July 2019), respectively. The versions of the maize genome sequence obtained from NCBI (https://www.ncbi.nlm.nih.gov/ (accessed on 1 July 2019) were used to identify ZmGSK proteins and their corresponding nucleotide sequences. All the *A. thaliana ASK* genes and encoded protein sequences, which were used as queries, were blasted against the NCBI database to search for GSK homologs in maize, with the E-value cutoff set as 1.00 × 10^−10^ and the coverage ratio as 50%. The Hidden Markov Model (HMM) profile of Pkinase (PF0069.24) was obtained from the Pfam protein families database (http://pfam.xfam.org/ (accessed on 28 July 2019) and used to verify maize GSKs. The candidate genes were further checked using pfamscan and the Pfam A database to ensure the presence of the Pkinase domain (PF0069.24).

### 4.3. Analysis of Chromosomal Localization

The maize GSK genes were located on the corresponding chromosomes using MapInspect software (https://mapinspect1.software.informer.com/ (accessed on 28 July 2019) according to the start positions indicated in the maize database.

### 4.4. Multiple-Sequence Alignment and Phylogenetic Analysis

The full-length amino acid sequences of GSK proteins from maize, rice, and Arabidopsis were aligned using MUSCLE (http://www.ebi.ac.uk/Tools/msa/muscle/ (accessed on 31 July 2019) and saved in the ClustalW format. Then, the unrooted phylogenetic tree was constructed using the neighbor-joining method in MEGA5.10 software with the bootstrap test replicated 1000 times.

### 4.5. Analysis of Promoter Cis-Regulatory Elements

To investigate the putative cis-acting elements of candidate *ZmGSK* genes, their promoter sequences (2000 bp upstream of the initiation codon “ATG”) were obtained with BLAST searches of the maize genome data using whole gene IDs. Potential cis-acting regulatory elements of the extracted sequences were subsequently examined using PlantCARE (http://bioinformatics.psb.ugent.be/webtools/plantcare/html/ (accessed on 13 August 2019) [53].

### 4.6. Analysis of Biophysical Properties of the GSK Genes

The maize GSK protein sequences were analyzed using the ExPASy-ProtParam tool (http://web.expasy.org/protparam/ (accessed on 17 August 2019) to calculate the number of amino acids, molecular weight, and theoretical pI. Meanwhile, the subcellular localization of these genes was predicted using ProComp 9.0 (http://www.softberry.com/berry.phtml?group=programs&subgroup=proloc&topic=protcomppl (accessed on 17 August 2019).

### 4.7. Analysis of Distribution of Conserved Domains and Exon–Intron Structure of ZmGSK Genes

MEME (http://meme-suite.org/) and SMATR (http://smart.embl-heidelberg.de/ (accessed on 24 August 2019) were used to elucidate the maize GSK domains, using the following parameters: motif width of 6–200 residues and maximum number of motifs =15. The mast.xml file exported from MEME was visualized using Tbtools_master (https://github.com/CJ-Chen/TBtools (accessed on 24 August 2019). Meanwhile, the intron/exon distribution was determined using the Gene Structure Display Server 2.0 program (http://gsds.cbi.pku.edu.cn/ (accessed on 24 August 2019).

### 4.8. Analysis of Expression Patterns of the GSK Genes

Expression data of the eleven maize *GSK* genes in different maize tissues were obtained from the MaizeGDB database. The expression patterns are presented as a heat map, which reflects log_2_(RPKM + 1), with red, white, and blue indicating high, medium, and low expression, respectively. Total RNA was extracted using TRIzol reagent (Invitrogen, Waltham, MA, USA). The isolated RNA (1.5 µg) was treated with DNase I, and cDNA synthesis was conducted using reverse-transcription PCR (TaKaRa, Dalian, China). The transcription levels of *ZmGSK* genes in various tissues were measured with semiquantitative RT-PCR, and *ZmGSK* gene expression levels following BR (10 nM) and ABA (100 μM) treatment of maize seeds (9–10 DAP) were measured using qRT-PCR. The primers used for semiquantitative RT-PCR and qRT-PCR are shown in Appendix A.

### 4.9. Yeast Two-Hybrid Assay

Yeast two-hybrid assays were performed as previously described. Full-length coding sequences of *ZmGSKs* were cloned into the pGBKT7 vector (BD) and transformed into yeast strain AH109 to test for auto-activation. Yeast on SD/-Trp and SD/-Trp-Ade-His agar plates was grown at 28 °C for 3 days. For the protein–protein interaction assay, ZmBZR1 was ligated to the pGADT7 vector (AD). Plasmid pGADT7-ZmBZR1 with pGBKT7-ZmGSKs were co-transformed into AH109 cells. Yeast strain AH109 was planted on SD/-Trp-Leu and SD/-Trp-Leu-Ade-His for 3 days. The primers used for cloned genes are shown in Appendix A.

### 4.10. Biomolecular Fluorescence Complementation (BiFC) Assays

The cDNA sequences of the *ZmGSK1,4,7,8,11* fragments were PCR-amplified and cloned into pSAT6-cEYFP, respectively. Full-length ZmBZR1-encoding sequences were inserted into pSAT6-nEYFP to generate an N-terminal fusion with nEYFP. For the BiFC protein–protein interaction analysis, the above constructs were introduced into onion with a helium biolistic gun transformation system. The bombarded samples were then cultured for from 24 to 48 h at 28 °C. Then, the samples were observed using BX61 fluorescent microscopy. The primers used for cloned genes are shown in Appendix A.

## 5. Conclusions

In summary, we identified and cloned 11 *GSK* genes in maize, which were distributed into four groups. We found that *GSK* genes are highly conserved among Arabidopsis, rice, and maize. Additionally, groups II and IV play important roles in the regulation of seed development by interacting with BZR1. Multiple lines of evidence suggest that ZmGSK1 plays a dominant role in BR signaling as the counterpart of BIN2. Overall, our results indicate that ZmGSK proteins are generally involved in the interaction with the ZmBZR1-mediated hormone pathway. Further, we show that some *GSK* gene members are involved in the BR, ABA, and BR–ABA hormone pathways.

## Figures and Tables

**Figure 1 ijms-23-08056-f001:**
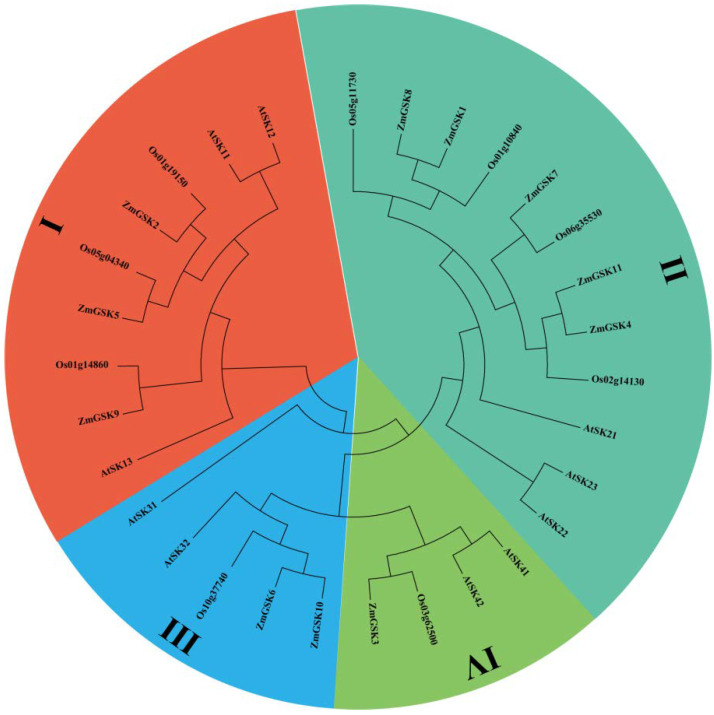
Phylogenetic-analysis results of GSK-3 proteins in Arabidopsis, rice, and maize. The phylogenetic tree was constructed using MEGA7 and the maximum likelihood method with 1000 bootstraps. Different clades of the GSK family are marked with different colors.

**Figure 2 ijms-23-08056-f002:**
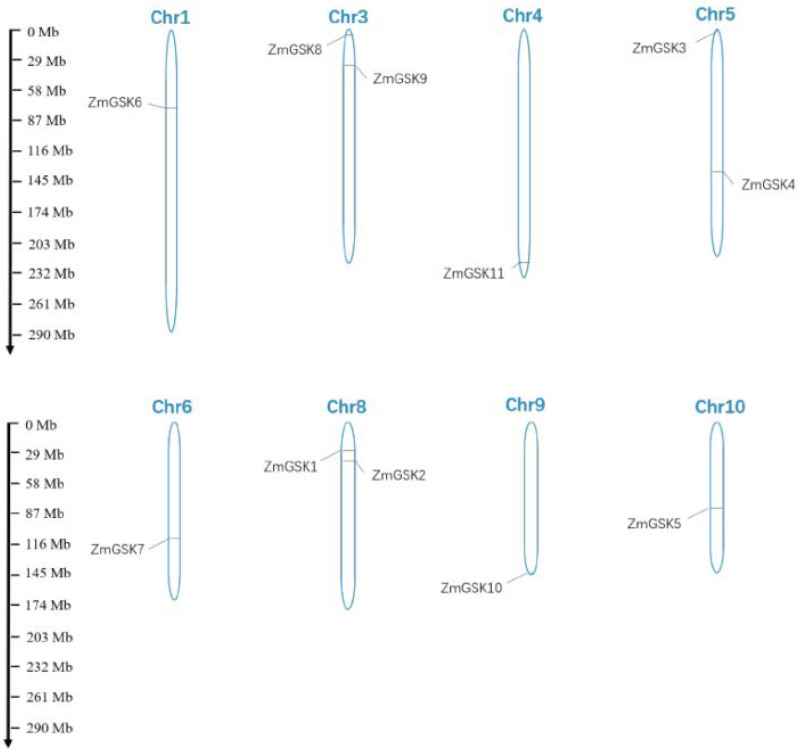
Distribution of ZmGSK genes on maize chromosomes. The scale represents megabases (Mb).

**Figure 3 ijms-23-08056-f003:**
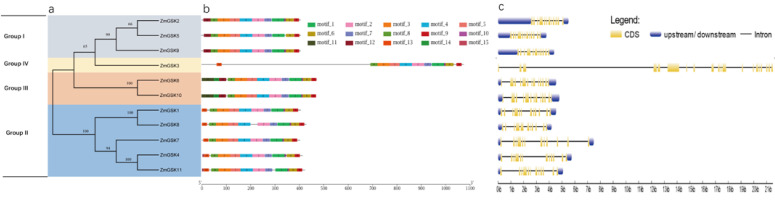
Phylogeny, exon–intron structures, and motif analysis of ZmGSKs. (**a**) Phylogenetic tree of ZmGSK. (**b**) Conserved motif analysis of ZmGSK OPF translations. Motifs were designated 1–10 and distinguished with different colors. (**c**) Exon–intron structures of *ZmGSK* genes. The full-length mRNA sequences of *ZmGSK* genes were analyzed and displayed. Coding sequences (CDSs) are represented by yellow boxes, untranslated regions (UTRs) are shown as blue boxes, and introns are represented by blank lines. The scale at the bottom represents lengths.

**Figure 4 ijms-23-08056-f004:**
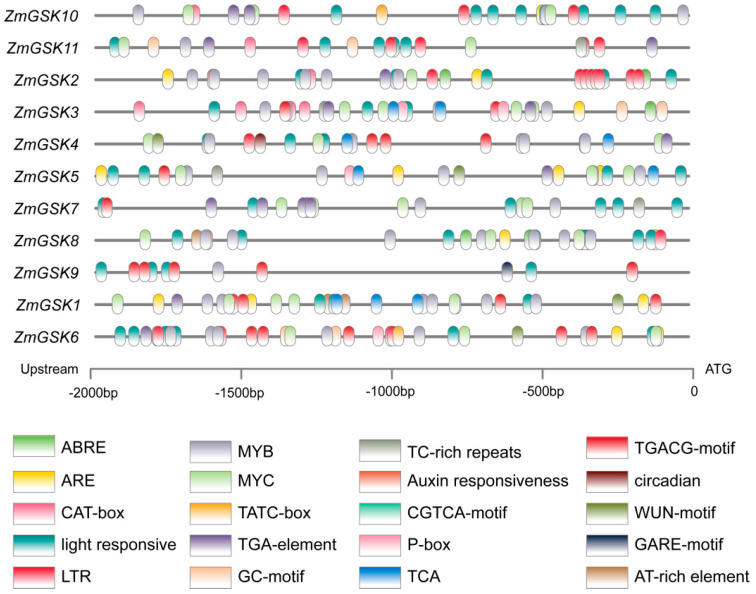
Predicted cis-acting regulatory elements in the promoter regions of GSK genes. Promoter sequences (−2000 bp) were analyzed with Plant CARE (Available online: http://bioinformatics.psb.ugent.be/webtools/plantcare/html/ (accessed on 13 August 2019).

**Figure 5 ijms-23-08056-f005:**
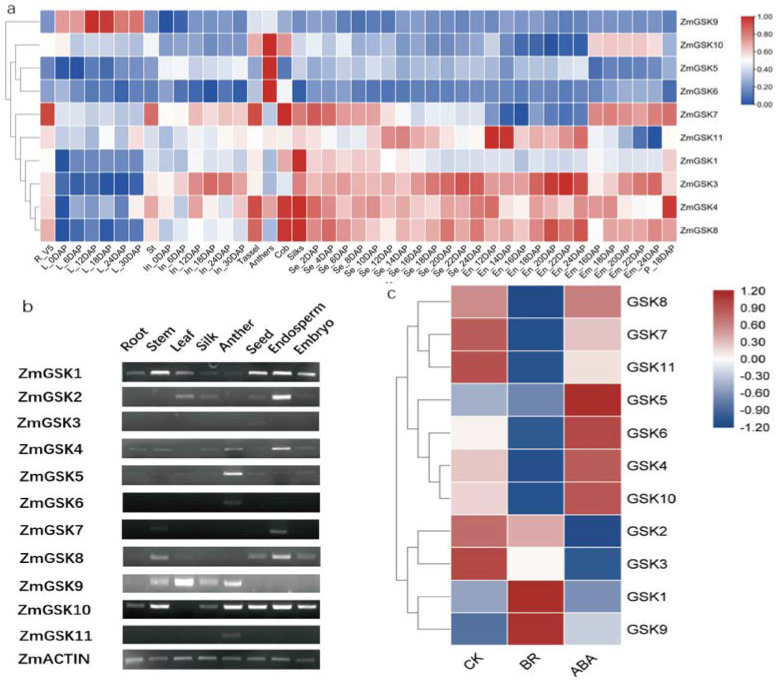
Expression patterns of *ZmGSK* genes. (**a**) Phylogenetic tree based on the MaizeGDB database. *ZmGSK2* is not in the tree because it was not found in the database. The tissues and developmental stages examined are indicated at the bottom of the heat map. The log2−transformed FPKM values were utilized to generate the heat map. (R: Root; L: Leaf; St: Stem; In: Internode; Se: seed; En: Endosperm; Em: Embryo) (**b**) Expression of *ZmGSK* genes in different tissues. (**c**) The relative expression levels of *ZmGSK* genes in 9−10 DAP seeds under normal, BR, and ABA conditions. The relative expression levels of the *ZmGSK* genes in comparison with that of the *ZmTXN* gene were used to draw the figure in TBtools software. The red and blue colors reflect log-transformed data with normalization.

**Figure 6 ijms-23-08056-f006:**
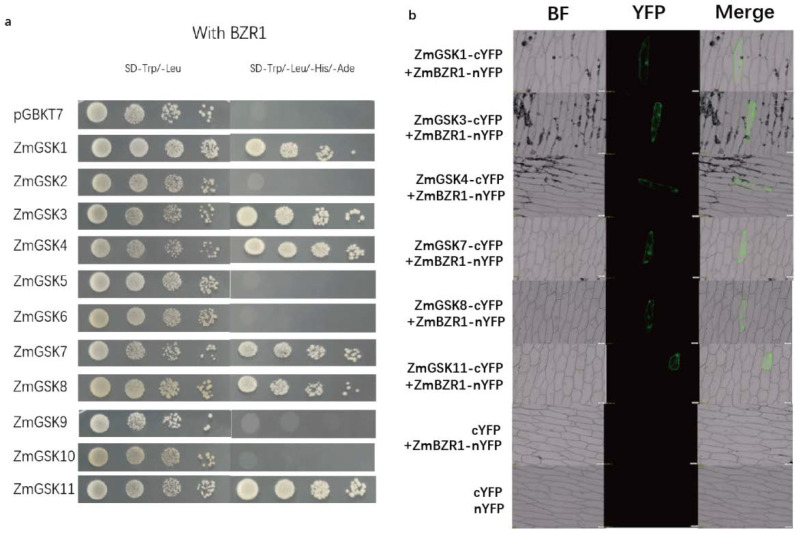
Results of protein–protein interaction assay of ZmGSKs and ZmBZR1. (**a**) Results of yeast two-hybrid assays. *Saccharomuces cerevisiae* strain AH109 was co-transformed with the indicated combinations of pAD-BZR1 and pBD-GSKs. The clones were first isolated on SD/-Trp/-Leu plates. Then, the transformed yeast colonies were grown on selection medium plates (SD/-Trp/-Leu/-His/-Ade). (**b**) BiFC-analysis results. Fluorescence was observed in the transformed cells, which resulted from the complementation of the C-terminal part of YFP fused with ZmGSKs (ZmGSKs–cYFP) with the N-terminal part of YFP fused with ZmBZR1 (ZmBZR1–nYFP).

**Table 1 ijms-23-08056-t001:** Basic information of GSK genes.

Subfamily	Gene ID	Proposed Name	Amino Acid Length	*M*_W_ (KDa)	PI	Location
Clade I	Zm00001d009055	ZmGSK2	406	46.2	8.39	8:33490549-33496057(+)
	Zm00001d024729	ZmGSK5	410	46.42	8.8	10:85508260-85512046(-)
	Zm00001d040263	ZmGSK9	408	46	8.9	3:34652668-34657191(+)
Clade II	Zm00001d008893	ZmGSK1	406	45.51	8.25	8:24167978-24172518(-)
	Zm00001d016188	ZmGSK4	412	46.02	8.59	5:149304159-149309955(+)
	Zm00001d037010	ZmGSK7	403	45.17	8.66	6:108958027-108965515(-)
	Zm00001d039407	ZmGSK8	429	47.99	7.95	3:3614006-3618190(+)
	Zm00001d053548	ZmGSK11	423	46.96	8.41	4:233637127-233642207(-)
Clade III	Zm00001d029664	ZmGSK6	473	53.02	6,79	1:81803033-81807676(-)
	Zm00001d048564	ZmGSK10	470	52.69	7.25	9:158839268-158844061(+)
Clade IV	Zm00001d012869	ZmGSK3	1076	120.83	6.33	5:1375911-1397930(+)

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
