# Peer review of "Genome-Wide Characterization and Phylogenetic Analysis of GSK Genes in Maize and Elucidation of Their General Role in Interaction with BZR1"

_ijms, 2022, doi:10.3390/ijms23158056_

Round 1
Reviewer 1 Report
This MS identified and characterized the GSK genes in maize as well as explored the potential interaction with BZR1. Overall, this paper is generally well-written and provides a fundamental data for further studies. However, there are some issues should be addressed.
1. L49 The full name of BIN2 should be provided.
2. What kind of software and method did you use for phylogenetic analysis? N-J method in MEGA 5.10 (L118-119 and L181) or maximum likelihood method in MEGA 7 (L191-192).
3. L182 and Fig 1. The tree has five clades instead of four clades (the clade III contains two distinct sub-clades). Did you name these clade based on previous publications? Besides, the current nomenclature of the GSK-3 family looks confusing. Consider providing gene ID of these sequences in supplemental files.
4. L182-183, in my opinion, it is very hard to conclude that this protein family may have conserved functions in plants, because there are many species-specific gene expansion events and it is barely has no one-to-one orthologous in the three species (Fig 1).
5. L275, The description of Fig 5 (b) is wrong.
6. L306-311. Consider rewriting or deleting these sentences.
7. L316-318. As mentioned, the number of GSK genes is different in the three species (10 GSK genes in Arabidopsis, 9 in rice and 11 in maize). Why is the number of GSK genes conserved among different plant genomes (L319)?
8. L 337-338, Base on your results, I think there are two types of BIN2 in maize (ZmGSK1 and ZmGSK8) and the duplication event happened after the divergence of the three species, because ZmGSK1 and ZmGSK8 have high similarity (Fig 1), tightly located in the chromosomal region (Fig 2) and presenting very similar expression pattern (Fig 5).
9. L359-360, I don’t think we can conclude one gene is more important than the other one, just simply based on the expression levels.
10. The resolution of images is too low.
Reviewer 2 Report
The manuscript identified 11 GSKs genes in maize genome, and found 5 GSKs genes in clads II may interact with ZmBZR1. The methods used in this manuscript is common, but author found ZmGSKs are important for BR signal, which is useful for the scientific researches. A small number of grammatical errors:
1. line 28 “Finally, Using the …” should be lowercase “using”.
2. line 47 it should “ZmGSK-3 proteins in maize”.
Reviewer 3 Report
Thank you for choosing me a potential reviewer for the manuscript entitled “Genome-wide characterization and phylogenetic analysis of GSK genes in maize and elucidation of their general role in in-teraction with BZR1” by Li et al.
I have reviewed this paper carefully and my suggestions and recommendations are as follow:
Please use the scientific name of the plant which you have been used such as rice and maize.
Please do not use the keywords which already mention in the title such as maize and BZR1.
In the introduction section, most of the citations are very old, please use the latest citations in your study.
“Maize is an important C4 crop with 80 high yield that contributes to solving the food security problem” Only this information is not enough for maize, please write some more details regarding the maize and cite these latest references: (Ali et al., 2022; Kamal et al., 2022; Saleem et al., 2022)
“Maize Research In-stitute of Sichuan Agricultural University.” Please also mention the latitude and altitude here.
Results are ok.
“Our findings will accelerate further functional characterization of ZmGSK genes and provide a framework to clarify the roles of these genes in biology and hormone response.” Please revise it.
“Overall, our results indicate that ZmGSK proteins are generally involved in the interaction with ZmBZR1 mediated hormone pathway.” Please mention some more future recommendations from your study.
References
Ali, B., Wang, X., Saleem, M.H., Sumaira, Hafeez, A., Afridi, M.S., Khan, S., Zaib-Un-Nisa, Ullah, I., Amaral Júnior, A.T.d., Alatawi, A., Ali, S., 2022. PGPR-Mediated Salt Tolerance in Maize by Modulating Plant Physiology, Antioxidant Defense, Compatible Solutes Accumulation and Bio-Surfactant Producing Genes. Plants 11, 345.
Kamal, A., Saleem, M.H., Alshaya, H., Okla, M.K., Chaudhary, H.J., Munis, M.F.H., 2022. Ball-milled synthesis of maize biochar-ZnO nanocomposite (MB-ZnO) and estimation of its photocatalyticability against different organic and inorganic pollutants. Journal of Saudi Chemical Society, 101445.
Saleem, M.H., Parveen, A., Khan, S.U., Hussain, I., Wang, X., Alshaya, H., El-Sheikh, M.A., Ali, S., 2022. Silicon Fertigation Regimes Attenuates Cadmium Toxicity and Phytoremediation Potential in Two Maize (Zea mays L.) Cultivars by Minimizing Its Uptake and Oxidative Stress. Sustainability 14, 1462.
